# c-Abl Tyrosine Kinase Is Regulated Downstream of the Cytoskeletal Protein Synemin in Head and Neck Squamous Cell Carcinoma Radioresistance and DNA Repair

**DOI:** 10.3390/ijms21197277

**Published:** 2020-10-01

**Authors:** Sara Sofia Deville, Luis Fernando Delgadillo Silva, Anne Vehlow, Nils Cordes

**Affiliations:** 1OncoRay—National Center for Radiation Research in Oncology, Faculty of Medicine Carl Gustav Carus, Technische Universität Dresden, 01307 Dresden, Germany; SaraSofia.Deville@uniklinikum-dresden.de (S.S.D.); anne.vehlow@nct-dresden.de (A.V.); 2Helmholtz-Zentrum Dresden—Rossendorf (HZDR), Institute of Radiooncology—OncoRay, 01328 Dresden, Germany; 3Center for Regenerative Therapies Dresden, Technische Universität Dresden, 01307 Dresden, Germany; luis.delgadillo@tu-dresden.de; 4National Center for Tumor Diseases, Partner Site Dresden, German Cancer Research Center, 69120 Heidelberg, Germany; 5German Cancer Consortium, Partner Site Dresden, German Cancer Research Center, 69120 Heidelberg, Germany; 6Department of Radiotherapy and Radiation Oncology, University Hospital Carl Gustav Carus, Technische Universität Dresden, 01307 Dresden, Germany

**Keywords:** ionizing radiation, HNSCC, synemin, c-Abl, DNA repair, zebrafish

## Abstract

The intermediate filament synemin has been previously identified as novel regulator of cancer cell therapy resistance and DNA double strand break (DSB) repair. c-Abl tyrosine kinase is involved in both of these processes. Using PamGene technology, we performed a broad-spectrum kinase activity profiling in three-dimensionally, extracellular matrix grown head and neck cancer cell cultures. Upon synemin silencing, we identified 86 deactivated tyrosine kinases, including c-Abl, in irradiated HNSCC cells. Upon irradiation and synemin inhibition, c-Abl hyperphosphorylation on tyrosine (Y) 412 and threonine (T) 735 was significantly reduced, prompting us to hypothesize that c-Abl tyrosine kinase is an important signaling component of the synemin-mediated radioresistance pathway. Simultaneous targeting of synemin and c-Abl resulted in similar radiosensitization and DSB repair compared with single synemin depletion, suggesting synemin as an upstream regulator of c-Abl. Immunoprecipitation assays revealed a protein complex formation between synemin and c-Abl pre- and post-irradiation. Upon pharmacological inhibition of ATM, synemin/c-Abl protein-protein interactions were disrupted implying synemin function to depend on ATM kinase activity. Moreover, deletion of the SH2 domain of c-Abl demonstrated a decrease in interaction, indicating the dependency of the protein-protein interaction on this domain. Mechanistically, radiosensitization upon synemin knockdown seems to be associated with an impairment of DNA repair via regulation of non-homologous end joining independent of c-Abl function. Our data generated in more physiological 3D cancer cell culture models suggest c-Abl as further key determinant of radioresistance downstream of synemin.

## 1. Introduction

DNA repair processes are fundamentally regulated by several extra-nuclear factors via not well-characterized mechanisms. Some examples of such regulators are growth factor receptors and integrin cell adhesion molecules. These proteins generate complexes, known as focal adhesions, which connect tumor cells to the extracellular matrix (ECM) and their cytoskeleton [1,2,3,4,5]. Several focal adhesion proteins (FAPs), such as β1 integrin, significantly confer tumor resistance to genotoxic agents by modulating repair of DSB [6,7,8,9,10]. In addition, accumulating evidence suggests that cytoplasmic FAP signaling is linked to nuclear repair dynamics through interactions between the pro-survival non-receptor tyrosine kinase c-Abl and components of the DNA repair machinery, including DNA-PKcs, ATM, BRCA1 and RAD51 [11,12]. These observations suggest a crosstalk between cytoskeletal factors and nuclear DNA repair-associated processes.

c-Abl is a well-known non-receptor tyrosine kinase, which regulates cell proliferation, differentiation, adhesion, migration and DNA-repair [13,14]. All these numerous functions promoted by c-Abl kinase signaling can be dysregulated during cancer. Therefore, c-Abl is emerging as potential target also in solid tumors [15,16] such as in melanoma, breast carcinoma and colorectal carcinoma [17,18,19]. c-Abl targeting strategies are becoming practical to overwhelm the over-activation of c-Abl as a result of genotoxic stress, growth factor receptor signaling or absent inhibitory stimuli [20].

Radiotherapy and various chemotherapeutics such as cisplatin induce DNA damage with DNA double strand breaks (DSB) being the most life-threatening subtype [21,22]. DSB can be repaired via two essential DNA repair mechanisms: homologous recombination (HR) and non-homologous end-joining (NHEJ). NHEJ repair is error-prone and is active during the whole cell cycle. Recently, we were able to demonstrate that FAP/cytoskeletal components such as the large type IV intermediate filament protein synemin influence radioresistance and DNA-PK regulated NHEJ in an ATM-dependent manner in head and neck cancer cells [23].

In the present study, we mechanistically demonstrate that synemin modulates c-Abl phosphorylation and activity and governs a fundamental role in DSB repair by serving as kinase anchoring protein for an ATM-dependent interaction with c-Abl. Taken together, our study provides evidence of cytoarchitectural elements such as intermediate filaments as key co-regulators of nuclear DNA repair.

## 2. Results

### 2.1. Synemin Regulates Radiochemosensitivity and DNA Double Strand Break Repair in HNSCC Cells

Previously, we reported that the intermediate filament synemin is involved in the response to X-ray irradiation in dependence on DNA-PKcs [23]. To further unravel the complex involvement of synemin in the regulation of radiation survival, we performed synemin knockdowns using three different siRNAs and one esiRNA (endoribonuclease-prepared siRNA) in two 3D lrECM grown HNSCC cell lines, i.e., Cal33 and SAS (Figure 1A–F and Appendix A). While basal clonogenic survival remained unaffected in both cell lines (Figure 1C,F), synemin silencing significantly enhanced radiosensitivity relative to controls (Figure 1B,C,E,F). In accordance with radiosensitization, we further addressed the function of synemin in DNA damage repair (Figure 1G–I) and upon cisplatin (CDDP) treatment alone or in combination with X-ray irradiation in synemin-depleted HNSCC cell cultures (Figure 1J–L). We found that synemin-silenced cells responded with significantly higher levels of residual 53BP1 foci when treated with 6 Gy X-ray exposure than controls (Figure 1H,I). Similarly, single CDDP exposure or combined CDDP/irradiation led to higher 53BP1 foci numbers under synemin silencing relative to controls (Figure 1K,L). Accordingly, numbers of residual γH2AX foci increased upon synemin silencing and irradiation relative to controls or when both treatments were combined (Figure 1G–L).

Due to the apparent involvement of synemin in the repair of radiochemotherapy-induced DSB, we characterized the subcellular distribution and expression of synemin without and in combination with irradiation. In immunofluorescence stainings of unirradiated cells, synemin predominantly localized to the cytoplasm with a marginal perinuclear accumulation and sparing of the cell membrane (Appendix A). Upon exposure to X-rays, the staining intensity and nuclear accumulation of synemin increased at early time points after irradiation along with elevated expression levels in whole cell lysates (for Cal33 cells already after 30 min, for SAS cells after 1 h post irradiation) (Appendix A). Similar results were observed in synemin expression kinetics using Western blotting (Appendix A).

### 2.2. Synemin Modulates Radiation Sensitivity in Zebrafish

Our results from more physiological 3D lrECM cell cultures prompted us to address the role of synemin in a more complex in vivo model. In this regard, zebrafish embryos are described as biosensor model system for cancer and treatment-related research (Figure 2A) [24]. Three readouts, i.e., cardiac edema, total body length and dorsal tail curvature, were analyzed after morpholino-mediated synemin knockdown and 10 Gy X-ray exposure (Figure 2B). Intriguingly, we found that synemin inhibition (Figure 2C) in combination with a 10 Gy X-ray irradiation significantly decreased zebrafish length (Figure 2D,F), significantly increased edema counts (Figure 2E), and led to an abnormal dorsal tail curvature (Figure 2D,G) as compared to wildtype control zebrafish. Thus, our in vivo analysis shows a radiosensitizing effect of synemin inhibition in zebrafish similarly to the observed effects in vitro. Collectively, our results suggest that synemin plays an essential role in cell survival.

### 2.3. Activity of c-Abl and Src Family Members Depends on Synemin

To further unravel how synemin elicits its impact on NHEJ, we performed a broad-spectrum kinase profiling. We found significant changes in the ability of tyrosine as well as serine/threonine kinases to phosphorylate specific peptide residues upon synemin silencing and X-ray irradiation after 1 h and 24 h relative to controls (Figure 3A–C and Appendix A). Next, we demonstrated that mostly all the tyrosine kinases showed a pronouncedly reduced kinase activity after synemin depletion (Figure 3A–C). The integrin signaling pathway, the EGF receptor signaling pathway as well as the cadherin signaling pathway were also highly downregulated 24 h post irradiation in synemin-depleted HNSCC cell cultures (Appendix A). Synemin knockdown alone failed to modify protein kinase activities (Figure 3A,B and Appendix A) implying that the DNA repair co-regulating function of synemin becomes essential upon genotoxic injury. Among the identified candidates, c-Abl turned out as top deactivated protein kinase along with members of the Scr family such as Scr, Yes, Lck, Blk, and Lyn (Figure 3A–C) as well as the ERBB family.

The observed dependency of c-Abl activity on synemin led us to dissect the underlying mechanisms. First, we examined c-Abl protein expression as well as its Y412 and T735 phosphorylation associated with c-Abl kinase activity. Upon synemin depletion and X-ray irradiation, both Y412 and T735 c-Abl sites demonstrated strongly reduced phosphorylation, while total c-Abl protein remained stable (Figure 3D,E). Moreover, overexpression of mCherry-Synemin revealed a stabilized c-Abl Y412 phosphorylation relative to mCherry controls indicative of a dependence of c-Abl activity on synemin (Appendix A). Taken together, our data suggest that synemin has a significant role in the regulation of c-Abl kinase activity potentially associated with the regulation of DNA damage repair.

### 2.4. Synemin Functionally Interacts with c-Abl

By means of colocalization analysis, we identified that synemin/c-Abl colocalization shows an increase after 1 h post irradiation followed by a decrease (Figure 4A and Appendix A). To investigate a potential physical interaction between synemin and c-Abl, we conducted immunoprecipitation (IP) assays with endogenous (synemin and c-Abl) and mCherry-Synemin in unirradiated and X-ray irradiated SAS cells. Strikingly, we found c-Abl bound to synemin relative to IgG and mCherry controls (Figure 4B–D and Appendix A).

Interestingly, c-Abl kinase is linked to DNA repair [12] and based on the prediction algorithm of Cytoscape [25] and of the group-based prediction system (GPS 3.0, [26]), c-Abl presents as central component of the NHEJ and HR repair machineries (Figure 4E). In order to elucidate the dependency of the synemin/c-Abl interaction on the kinase activity of c-Abl or the kinase activities the two key DNA repair enzymes ATM and DNA-PKcs, we pharmacologically inhibited c-Abl (Imatinib), DNA-PKcs (NU7026) and ATM (KU55933) prior to immunoprecipiting synemin. We found reduced binding of c-Abl and DNA-PKcs in ATMi-treated, unirradiated cells as well as DNA-PKcsi- or ATMi-treated, irradiated cells (Figure 5F and Appendix A). To better understand if synemin interacts with the ΔSH2 domain of c-Abl, we generated a c-Abl construct deleted in the SH2 domain (Figure 4G). Upon immunoprecipitation, we observed that the interaction between synemin and c-Abl was partially lost (Figure 4H).

### 2.5. Synemin Influences DSB Repair and Radiation Survival Responses by Regulating c-Abl

To underpin the interdependency of synemin and c-Abl, we quantified clonogenic survival after single and double knockdown without and in combination with irradiation (Figure 5A–C). While basal cell survival was unaffected by most conditions (Figure 5A,B), single synemin knockdown significantly enhanced cellular radiosensitivity relative to controls and c-Abl silencing (Figure 5A,C). In line with these results, double synemin/c-Abl knockdown induced radiosensitization to the same extend as single synemin knockdown (Figure 5C).

We next investigated the functional relationship between synemin and c-Abl for DSB repair. Single and double depletion of synemin and c-Abl intriguingly revealed similar residual γH2AX and 53BP1 foci numbers relative to controls (Figure 5D,E). This suggests that (i) synemin critically impacts on the functionality of c-Abl in DSB repair and (ii) synemin and c-Abl are components of the same signaling pathway.

Owing to the fact that the synemin/c-Abl interaction seems to be controlled by non-homologous end joining enzymes, we subsequently explored NHEJ activity using a reporter assay upon single and double synemin and c-Abl depletion (Figure 5F). We observed an approximate 50% reduction in synemin- and synemin/c-Abl-depleted HNSCC cell cultures in contrast to controls and an approximate 50% increase in NHEJ activity in c-Abl-depleted cell cultures (Figure 5G). Thus, our results suggest a dependency of the DNA damage response on the functional interaction of c-Abl and synemin.

## 3. Discussion

Extra- and intracellular players orchestrate radio- and chemoresistance of cancer cells through complex and yet not fully understood mechanisms. The intermediate filament synemin, also defined as focal adhesion protein, has been shown to confer radioresistance by co-regulating DNA-PKcs-dependent DNA repair processes [23]. Here, we further elucidated the involvement of synemin in radioresistance and DNA repair and add additional facets. We demonstrate that (i) reduced activity of 86 tyrosine kinases, including c-Abl, in synemin-depleted, irradiated HNSCC cell cultures, (ii) c-Abl hyperphosphorylations on Y412 and T735 upon irradiation are significantly diminished, (iii) simultaneous synemin/c-Abl targeting elicits radiosensitization and DSB repair superimposable to single synemin depletion, (iv) synemin and c-Abl form a protein complex dependent on ATM activity, (v) deletion of the ∆SH2 domain in c-Abl is critical for synemin interactions with c-Abl, (vi) synemin knockdown decreases non-homologous end joining activity, and (vii) experiments using an in vivo zebrafish model confirm synemin as novel crucial determinant of cellular radiosensitivity.

While more and more enzymes and co-factors originally considered part of DNA repair mechanisms are found outside the nucleus, transmembrane and cytoplasmic molecules are apparently identified as important co-regulators of the DNA damage response and DSB repair [27,28]. In addition to the here described IF protein synemin, Ku proteins, ATM and DNA-PK have been documented to participate in cell adhesion, migration, etc. [29,30]. Based on synemin’s focal adhesion and cytoplasmic localization as well as its scaffold function, it is not surprising how kinase activities were modified in the absence of this protein. However, the great selectivity for tyrosine kinases was just as astonishing as the fact that the full impact of synemin depletion only became apparent in genotoxically injured cells. The kinome dataset generated in irradiated, synemin-depleted cell cultures showed an overall decrease in tyrosine kinase activity, including several essential kinases involved in pro-survival mechanisms, such as Scr, Yes, Lck, Blk, and Lyn, as well as ERBB2, 3 and 4. In the overall signaling pathway analysis, the integrin signaling pathway, the EGF receptor signaling pathway as well as the cadherin signaling pathway were also highly downregulated 24 h post irradiation in synemin-depleted cells. As previously shown by us and others, integrin and EGF receptor targeting, including some of their downstream signaling mediators, effectively confer sensitization to radio(chemo)therapy [2]. This is in line with the radiosensitizing effects observed upon synemin knockdown. This observation warrants further in-depth investigations hypothesizing a contribution of synemin to proper signaling in many survival advantaging signaling axis.

Interestingly, one of the most downregulated kinases upon synemin depletion was c-Abl. This was in line with the changes in phosphorylation of the amino acid residues Y412 and T735 of c-Abl. Due to the known role of c-Abl in DNA repair processes [12,31,32], we investigated whether c-Abl tyrosine kinase is an additional important mediator integrating synemin signals into radioresistance. Here, we demonstrate radiosensitization in 3D lrECM HNSCC cell cultures simultaneously depleted of synemin and c-Abl, which was superimposable with the degree of radiosensitization observed after single synemin targeting. Interestingly, c-Abl deactivation was markedly less effective regarding sensitization to irradiation. By using DNA repair reporter and DNA repair foci assays, we generated corroborating data.

Intriguingly, these findings about synemin/c-Abl interactions are also consistent with a previous report demonstrating interactions of synemin and DNA-PK to co-control DSB repair [23]. Several publications document a complex signaling network between c-Abl and different DNA repair proteins. The tyrosine kinase activity of c-Abl is induced during the DNA damage response subsequently phosphorylating DNA repair proteins like ATM and DNA-PKcs [11,33]. In general, c-Abl’s enzymatic activity is regulated through phosphorylation on Y412 [20]. Interestingly, we found higher levels of phospho-c-Abl Y412 upon X-ray exposure relative to un-irradiated cells. However, when Synemin was inhibited, the phospho-c-Abl Y412 dynamics was opposite to that of the control and showed a significantly weaker expression. During the DNA damage response, c-Abl dissociates from cytoplasmic 14-3-3 proteins and shuttles into the nucleus to interact with various DNA repair proteins to promote DNA repair [33,34,35]. While the cytoplasmic-to-nuclear shuttling was not addressed in our study, we observed synemin accumulation in the perinuclear area as well as inside the nucleus post X-ray exposure. In our experiments in synemin-depleted cell cultures, we also observed changes in the phosphorylation of the amino acid residue T735 of c-Abl, a phosphorylation determining the localization of c-Abl in the cell. We like to speculate that this mechanism assists the DNA repair and survival proficiency of normal and cancer cells. This idea is supported by the fact that synemin interacts with c-Abl independent of X-ray exposure. Nevertheless, futures studies are needed to address the functional consequences of this finding.

To address c-Abl´s interrelation with other DNA repair factors, we focused on ATM as putative candidate. In a recent study, synemin has been shown to functionally interconnect with DNA-PK via ATM [23]. Interestingly, our data support the notion that ATM is associated with synemin and c-Abl as ATM inhibition does affect their protein-protein interaction. To further evaluate the interaction between synemin and c-Abl, we used a wildtype c-Abl and a c-Abl ΔSH2 deletion construct. Here, we found that the interaction between these proteins is partially lost in the absence of the SH2 domain generally inevitable for proper protein-protein binding. Reasons for detecting only a partial and not a total loss of binding might lie in the presence of endogenous wildtype c-Abl. Linking these observations to c-Abl kinase activity as well as synemin/c-Abl signaling requires further experiments. Nonetheless, these data further corroborate our notion of the synemin/c-Abl interaction as an important determinant of radioresistance in HNSCC cell lines. Future investigations are necessary to untangle the molecular circuitry involving intermediate filaments, tyrosine kinase signaling and DNA repair.

## 4. Materials and Methods

### 4.1. Cell Lines and 3D Cell Culture

HNSCC cell lines (Cal33 and SAS) were kindly provided by R. Grenman (Turku University Central Hospital, Turku, Finland). Cal33-pimEJ5GFP were kindly provided by K. Borgmann (University Medical Center Hamburg-Eppendorf, Hamburg, Germany). Cells were asynchronously grown in Dulbecco’s modified Eagle’s medium containing glutamax-I (from AppliChem, Darmstadt, Germany) supplemented with 10% fetal calf serum and 1% non-essential amino acids (all from PAA Laboratories, Cölbe, Germany) at 37 °C in a humidified atmosphere containing 8.5% CO_2_. For 3D cell cultures, cells were embedded in 0.5 mg/mL laminin-rich extracellular matrix (lrECM) (Matrigel; BD, Heidelberg, Germany) as described previously [23]. All cell lines were authenticated using a STR DNA profiling and tested negative for mycoplasma contamination.

### 4.2. X-ray Irradiation

Irradiation was performed at room temperature using single doses of 200-kV X-rays (Yxlon Y.TU 320; Yxlon; dose rate ~ 1.3 Gy/min at 20 mA) filtered with 0.5 mm Cu as described previously [23]. Dosimetry for quality assurance was performed using a Duplex dosimeter (PTW, Freiburg, Germany) prior to irradiation.

### 4.3. Antibodies

Antibodies against c-Abl (#2862P), c-Abl T715 (#2846S), and c-Abl Y412 (#2865S) were from Cell Signaling (Frankfurt, Germany); β-actin (#A1978) and Synemin (#S9075, for zebrafish western blot) were from Sigma-Aldrich (Taufkirchen, Germany); γH2AX S139 (#05-636) was from Upstate (Schwalbach, Germany) and 53BP1 (#NB100-904) was from Novus Biologicals (Cambridge, UK). Desmuslin for western blots (#ab204369) and mCherry (#ab183628) were from Abcam (Cambridge, UK). Desmuslin (#sc-374484) for immunoprecipitation and immunofluorescence was from Santa Cruz (Heidelberg, Germany). Horseradish peroxidase-conjugated donkey anti-rabbit (#NA-934) and sheep anti-mouse (#NA-931) secondary antibodies were from GE Healthcare (Solingen, Germany); horseradish peroxidase-conjugated donkey anti-rabbit (#GTX221666-01) and sheep anti-mouse (#GTX221667-01) secondary antibodies for immunoprecipitation were from GeneTex Irvine (Irvine, CA, USA). Alexa Fluor 594 anti-mouse (#A11032), Alexa Fluor 594 anti-rabbit (#A11037), Alexa Fluor 488 anti-mouse (#A11029), Alexa Fluor 488 anti-rabbit (#A11034), and Alexa Flour 594 Phalloidin (#A12381) were from Life Technologies GmbH (Karlsruhe, Germany).

### 4.4. esiRNA and siRNA Transfection

Endoribonuclease-prepared siRNA (esiRNA) for Synemin (esiSYNM, 5′-AAACAGACCAGAAACCATCCGAACAAAGCCAGAA-GAGAAAATGTTCGATTCTAA-3′) and relative RLUC control (esiCTRL, 5′-ATTTATTAATTATTATGATCAGAAAAACATGCAGAAAATGCTGTTATTTTTTTAC-3′) were purchased from Eupheria Biotech (Dresden, Germany). siRNA for siSYNM#1 (5′-GCCGAUUAGUCUAGAAGUAtt-3′), siSYNM#2 (5′-CGGUGAAUUUCAUGCCGAAtt-3′), siSYNM#3 (5′-GCCUUACCAUGCAUUUCCGtt-3′), sic-Abl (5′-GGCCAUCAACAAACUGGAGtt-3′) and Silencer Negative Control siRNA (AM4635, siCTRL, 5′-AAAACAGUUGCGCAGCCUGAAtt-3′) were obtained from Ambion (Darmstadt, Germany). siRNA transfection was carried out as published in [23]. In brief, 24 h after plating, 26 nM/20 nM of esiRNA/siRNA was delivered using 8 µL/4 µL oligofectamine respectively and Opti-MEM (Invitrogen, Karlsruhe, Germany) under serum-free condition for 8 h. Subsequently, Opti-MEM plus 10% FCS was added to the cells. Twenty-four hours post transfection, cells were used for performing DNA DSB and clonogenic survival assays.

### 4.5. Total Protein Extraction, Western Blotting

Twenty-four hours after transfection with esiRNA (esiCTRL and esiSYNM), cells were re-seeded in 3D lrECM. Next day, cells were irradiated with 6 Gy X-rays or left unirradiated. The harvesting of total cell lysates was performed after 0.5, 1, 2, 6 or 24 h post irradiation. Whole cell lysates, SDS-PAGE and Western blotting were performed as previously described in [23]. In brief, cells were lyzed with lysis buffer supplemented with protease inhibitor (Complete protease inhibitor cocktail from Roche) and phosphatase inhibitors (Na_3_VO_4_ and NaF from Sigma, Taufkirchen, Germany). The lysates were then incubated for 30 min on ice; the cell membranes where then broken using a syringe and 1 h later centrifuged at 13,000× *g* for 20 min to remove debris. Since in 3D lysates it is not possible to quantify protein levels with common protein quantification kits, a gel for β-actin was necessary to evaluate the protein levels. After β-actin quantification using imageJ, proper dilutions were prepared. The chemiluminescent detection was preformed using ECL™ Prime Western Blotting System (Sigma-Aldrich, Taufkirchen, Germany).

### 4.6. 3D Colony Formation Assay

The 3D colony formation assay was applied for measuring clonogenic cell survival as published [23]. Cells were transfected with esiRNA/siRNA (see esiRNA and siRNA transfection) and next day embedded into 0.5 mg/mL lrECM in 96-well plates. Cells were irradiated 24 h post reseeding and kept for several days (cell line-dependently) in 8.5% CO_2_ at 37 °C. Each point on the survival curve represents the mean surviving fraction from at least three independent experiments.

### 4.7. Foci Assay

For determination of DSBs, cells were stained for γH2AX and 53BP1 as described in ref [23]. Briefly, 24 h after irradiation, cells were isolated using trypsin/EDTA, fixed with 3% formaldehyde/phosphate-buffered saline (Merck, Darmstadt, Germany) and permeabilized with 0.25% Triton-X-100/phosphate-buffered saline (Roth, Karlsruhe, Germany). Staining was accomplished with specific antibodies and Vectashield/4′-6-diamidino-2-phenylindole (Alexis, Lörrach, Germany) was used as mounting medium. Foci were counted microscopically with an AxioImager A1 plus fluorescence microscope (Carl Zeiss, Jena, Germany) under a ×40 objective. Immunofluorescence images were sustained using LSM 510 meta (Carl Zeiss) or AxioImager M1 (Carl Zeiss). For testing the impact of chemotherapy and radiochemotherapy, cells were transfected with esiRNA and next day embedded into 0.5 mg/mL lrECM in 24-well plates. Twenty-four hours after 1.7 µM of Cisplatin or DMSO was added to the cells. After 1 h cells were exposed to 6 Gy X-rays or left untreated. On the next day, cells were fixed and stained for residual foci. The staining was performed as published [23]. Three independent experiments were performed and 50 cells were quantified per condition/trial.

### 4.8. Synenim Constructs and Stable Transfection

Human mCherry-Synemin was kindly provided by R. J. Bloch (University of Maryland, USA) and c-Abl WT mouse cDNA was kindly provided by A. Koleske (Yale University, New Haven, CT, USA) and M. Krause (King’s College London, London, UK) [36]. Different constructs were generated using PCR. Set of primers for ΔSH2 deletion are the following: ABL1-ΔSH2-F (GGTGTGAAGCCCAAACGAAAAATCTGTACGTGTCCTCCGAGAGCCG-F) and ABL1-ΔSH2-R (CGGCTCTCGGAGGACACGTACAGATTTTTCGTTTGGGCTTCACACC-R). Set of primers were purchased from MWG Eurofins (Ebersberg, Germany). The primers used for the generation of all the constructs were designed with HindIII restriction site for the forward primer and BamHI restriction site for the reverse primer. Stable transfection of the synemin constructs was performed as published [23] using lipofectamine2000 (Invitrogen, Karlsruhe, Germany) and G418 (#A1720, Sigma-Aldrich, Taufkirchen, Germany) for selection of cells.

### 4.9. Zebrafish Lines and Maintenance

Zebrafish WT AB were used in the experiments. Zebrafish were raised under standard conditions at 28 °C. Embryos for microinjection were obtained from crossing WT AB zebrafish. Experimental manipulations were performed with larvae less than 5 days post fertilization as stated in the animal protection law (TierSchVersV §14). According to the EU directive 2010/63/EU, the use of these earlier zebrafish stages reduces the number of experimental animals, according to the principles of the 3Rs.

### 4.10. Analysis of Zebrafish Embryos

For determination of zebrafish radiosensitization upon synemin knockdown, zebrafish embryos were injected with 1 or 2 ng synemin-specific or control morpholino. The synemin (moSYNM: CGGTTCCCTCATTCGAAACATTTCCCCG) and mismatch control (moCTRL: CGGTCCTCACTTTAGCAATACTCCTCCG) morpholinos (GENETOOLS, LLC, Philomath, OR, USA; www.gene-tools.com) were designed to target the translational start site (ATG). Morpholinos were dissolved in PBS with 0.01% phenol red. Fertilized 1-cell stage embryos were microinjected in the yolk with 1 nL of morpholinos with either control (moCTRL) or synemin (moSYNM) at a concentration of 1 ng or 2 ng. Injected embryos were transferred to 100 mm dishes and were incubated at 28 °C in E3 medium (5 mM NaCl, 0.17 mM KCl, 0.33 mM CaCl_2_, 0.33 mM MgSO_4_, 0.0005 methylene blue) until further use. Unfertilized eggs were removed from the dishes and the remaining embryos were irradiated at 24 h.p.f. At 24 h.p.IR, underdeveloped embryos were removed. Zebrafish larvae (<5 d.p.f.) were scored for dorsal tail curvature and morphological changes (length and edema). Pictures were obtained of tricaine-anaesthetized larvae mounted on 1% low-melting agarose on a 96-well plate and imaged with Celigo Imaging Cytometer.

### 4.11. pimEJ5GFP-Based Chromosomal Break Reporter Assay

For measuring NHEJ activity, the pimEJ5GFP plasmid were stably transfected to generate an isogenic cell line pair (Cal33) as published in [23,37]. To measure NHEJ-mediated repair, cells were transiently transfected with pcDNA3BMyc-NLS-ISceI to express the I-SceI endonuclease for DSB induction [38]. Along with I-SceI, the pEGFP-N1 plasmid (Clontech, Mountain View, CA, USA), was transfected for determining transfection rates. Transfection was performed using lipofectamine2000 according to the manufacturer´s protocol. At 8 h after esiRNA transfection, cells were transfected with I-SceI and pN1 plasmids. Four hours thereafter, cells were trypsinized and were then reseeded in 3D (into 0.5 mg/mL lrECM). At 72 h cell were trypzinized, filtered and subjected to flow cytometry (FACS Canto; BD Biosciences, Beckton, Dickinson). Per sample, 2 × 10^4^ events were measured. GFP-positive cells were normalized to pEGFP-N1-positive cells and analysis was performed using FlowJo software (version 7.6.2).

### 4.12. Kinome Analysis

PamChip peptide microarrays, STK and PTK, were purchased from PamGene (BJ’s-Hertogenbosch, The Netherlands). Each array contained 140 (STK) and 144 (PTK) peptides with known phosphorylation sites. After synemin depletion using esiRNA, 1.5 × 10^6^ cells were plated into 3D lrECM, and irradiated with 6-Gy 24 h after. Cells were lyzed with 10× kinase buffer (#9802, Cell Signaling, Frankfurt, Germany), supplemented with HALT phosphatase and protease inhibitor cocktail (#1862495 and #1862209, Thermo Scientific, Darmstadt, Germany) at 1 and 24 h post irradiation. The lysates were then incubated for 30 min on ice and centrifuged at 13,000× *g* for 20 min to remove debris. Snap-frozen samples in triplicate were sent to the Genomics and Proteomics Core Facility Microarray Unitcenter at DKFZ (Heidelberg, Germany) for analysis of kinase activity. In brief, sample incubation, detection, and analysis were performed in a PamStation 12 system according to the manufacturer’s instructions. In brief, the arrays were blocked with 2% BSA in water for 30 cycles and washed 3 times with PK assay buffer. Kinase reactions were performed for 1 h at 30 °C. Phosphorylated peptides were detected using fluorescence labeled antibodies (anti-rabbit–FITC) that recognize specific antibodies against phosphorylated peptides. FITC-labeled arrays were then imaged using a 12-bit CCD camera. BioNavigator software (PamGene International BV) was employed for quantification of the images obtained from the phosphorylated arrays. A list of significantly phosphorylated peptides was generated from control and knockdown samples and analyzed with GeneGo, PhosphoSite database and KinMap to predict main down/upregulated kinases.

### 4.13. Immunoprecipitation

For precipitation of antibody-protein complex out of a cell lysate, protein A/G-beads (#PRAG25-A5-5, Alpha Diagnostics Intl., San Antonio, TX, USA) were used as described in ref [23]. In short, 4.5 × 10^6^ SAS cells stably transfected with mCherry-C1 and mCherry-Synemin were harvested using cell lysis buffer (Cell Signaling, Frankfurt, Germany) supplemented with 40 µL/mL Complete protease inhibitor cocktail. The total protein amount was measured by BCA assay. Cell lysates were pre-cleared using 50 μL of Protein A/G sepharose slurry (50% *v*/*v*). To do this, the lysate-bead solution was rotated at 4 °C for 1 h using a laboratory rotator. Following pre-clearing, lysates were centrifuged at 500× *g* for 5 min and the supernatant was transferred to a new reaction tube. Primary antibodies (IgG as isotype control) were added to 1 mg protein lysate and rotated for 1 h at 4 °C. Subsequently, 50 μL of Protein A/G sepharose slurry (50% *v*/*v*) was added and rotated overnight at 4 °C. Immunoprecipitates were washed once with 600 µL of ice cold lysis buffer. Whole cell lysates and immunoprecipitated proteins were boiled in 50 μL sample buffer, separated by SDS-PAGE, transferred, and blotted. Protein precipitates were analyzed with specific primary antibodies as indicated previously.

### 4.14. Data Analysis

Means ± standard deviation (SD) of at least three independent experiments were calculated with reference to controls defined in total numbers or 1.0. For statistical significance, two-sided Student’s *t*-test was performed using Microsoft Excel 2016 or one-way ANOVA followed by post-hoc analysis using Tukey’s correction was carried out by using Prism7 (GraphPad, San Diego, CA, USA). *p*-value of less than 0.05 was considered statistically significant. For zebrafish experiments One-way ANOVA followed by post-hoc analysis using Tukey’s correction was used. Logistic regression analysis were performed using SPSS.

## 5. Conclusions

DNA repair appears to be more complex than expected and our observations add further extracellular facets that should be considered in the future with regard to the regulation of DSB repair. In summary, our data show how the intermediate filament Synemin together with c-Abl controls the radiation sensitivity and DNA repair of HNSCC cells. These interactions between Synemin, c-Abl and ATM further underline the concept of cytoarchitectonic elements as important co-regulators of nuclear events.

## Figures and Tables

**Figure 1 ijms-21-07277-f001:**
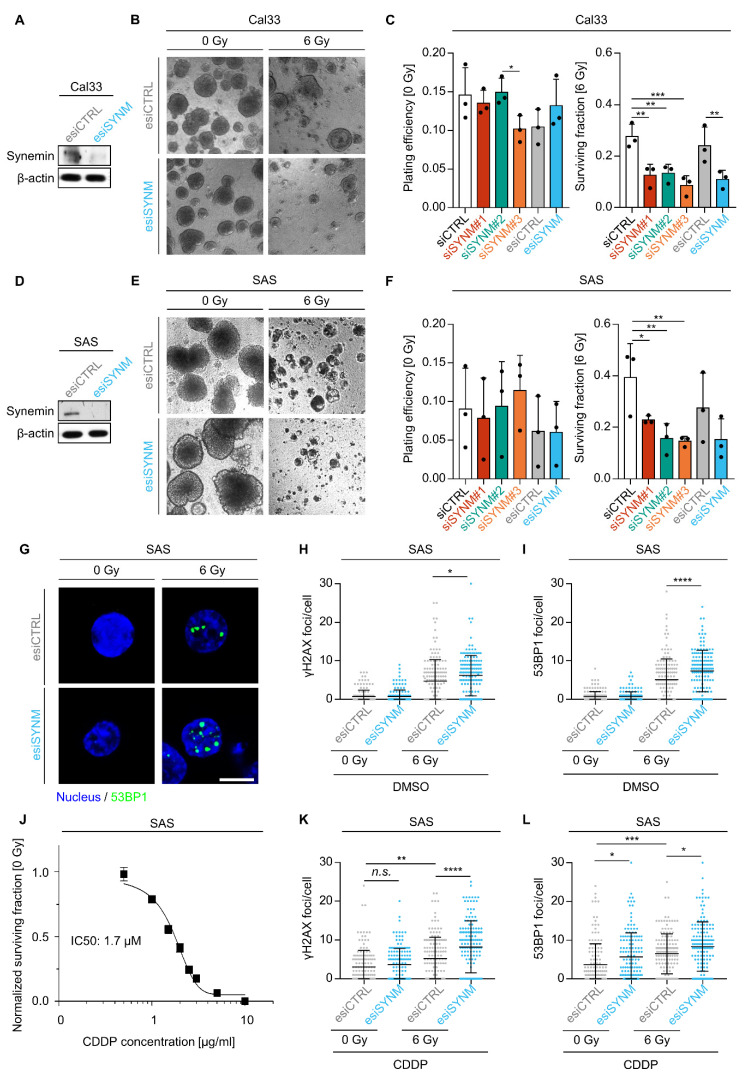
Synemin essentially controls cellular radiochemosensitivity of 3D lrECM HNSCC cell cultures. (**A**) Knockdown efficiency of control and synemin-specific esiRNA in whole cell lysates from Cal33 cells. (**B**) Representative phase contrast images of 3D lrECM Cal33 cell cultures (bar, 200 µm). (**C**) Plating efficiencies of un-irradiated Cal33 cells and surviving fraction of Cal33 cells after 6 Gy X-ray exposure (*n* = 3). Dots represent the mean of each independent experiment. (**D**) Knockdown efficiency of control and synemin-specific esiRNA in whole cell lysates from SAS cells. (**E**) Representative phase contrast images of 3D lrECM SAS cell cultures (bar, 200 µm). (**F**) Plating efficiencies of un-irradiated SAS cells and surviving fraction of SAS cells after 6 Gy X-ray exposure (*n* = 3). Dots represent the mean of each independent experiment. Data are presented as mean ± SD (two-sided *t*-test; * *p* < 0.05, ** *p* < 0.01, *** *p* < 0.001). (**G**) Representative immunofluorescence images of 53BP1 foci (green) and nucleus (DAPI, blue) (bar, 10 µm). (**H**) γH2AX and (**I**) 53BP1 residual foci numbers upon synemin inhibition plus/minus X-ray exposure. (**J**) Dose-response relationship of SAS cells treated with increasing Cisplatin (CDDP) concentrations (*n* = 3; IC50, inhibitory dose at 50% survival). (**K**) γH2AX and (**L**) 53BP1 foci numbers upon synemin inhibition and CDDP treatment in combination with and without 6-Gy irradiation (*n* = 3; at least 50 cells were quantified per condition per trial). Data are presented as mean ± SD (one-way ANOVA followed by post-hoc analysis using Tukey’s correction; * *p* < 0.05, ** *p* < 0.01, *** *p* < 0.001, **** *p* < 0.0001).

**Figure 2 ijms-21-07277-f002:**
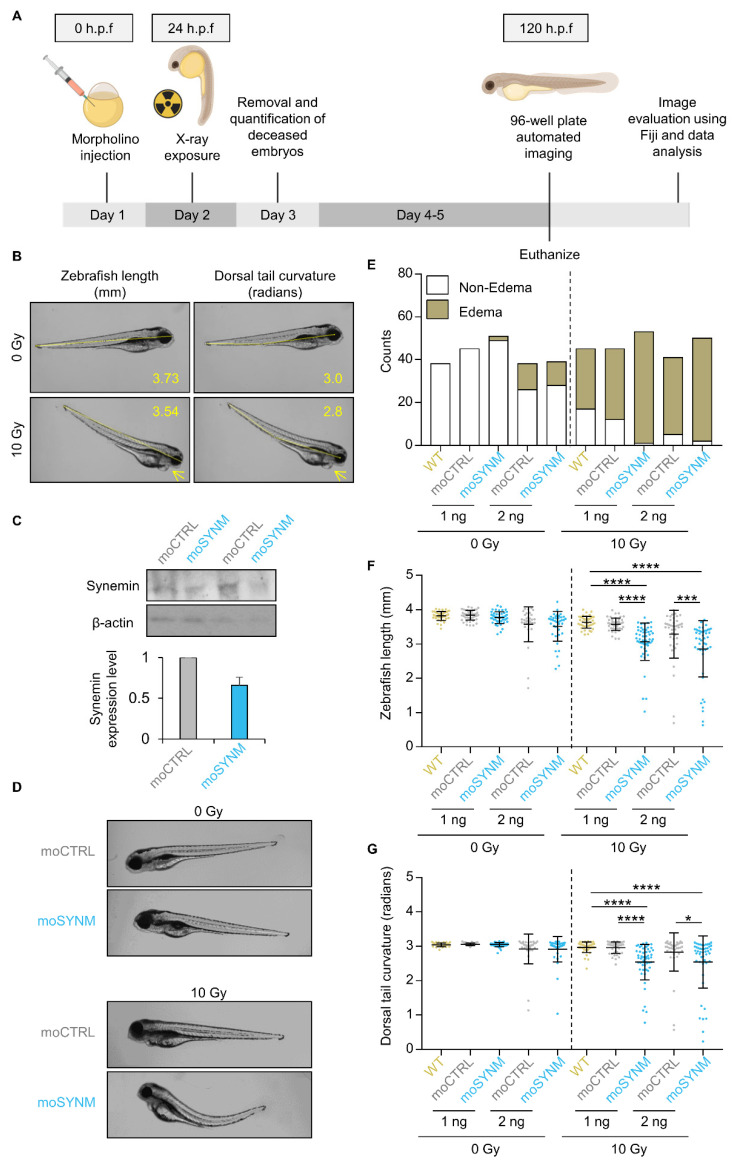
Targeting synemin elicits enhanced radiosensitivity in zebrafish. (**A**) Workflow of experimental set up. Immediately after fertilization (0 h post fertilization (h.p.f.), morpholinos were injected with different concentrations (1 or 2 ng). Scrambled non-specific control morpholino (moCTRL) or synemin-specific morpholino (moSYNM) were applied or zebrafish were left untransfected (wild type (WT)). Next day, the embryos were irradiated with 10 Gy X-ray. Zebrafish larvae were imaged for quantification of morphological changes and subsequently euthanized prior to 120 h.p.f. (**B**) Representative images of zebrafish larvae upon indicated treatment. Edema, length, and dorsal tail curvature were quantified using Fiji software. Numbers in yellow indicate the lengths measurements and the yellow arrows indicate edema. (**C**) Knockdown efficiency after control and synemin-specific morpholino transfection in whole cell lysates from zebrafish. Representative Western blots and densitometries are shown. (**D**) Representative images of zebrafish larvae tail curvature upon indicated treatment. (**E**) Quantification of edema upon indicated treatments (*p*-value, logistic regression analysis). (**F**) Quantification of zebrafish length and (**G**) dorsal tail curvature for indicated treatment conditions. Data are presented as mean ± SD (*n* = 2; at least 15 embryos per trial were evaluated, N > 30; one-way ANOVA followed by post-hoc analysis using Tukey’s correction; * *p* < 0.05, *** *p* < 0.001, **** *p* < 0.0001).

**Figure 3 ijms-21-07277-f003:**
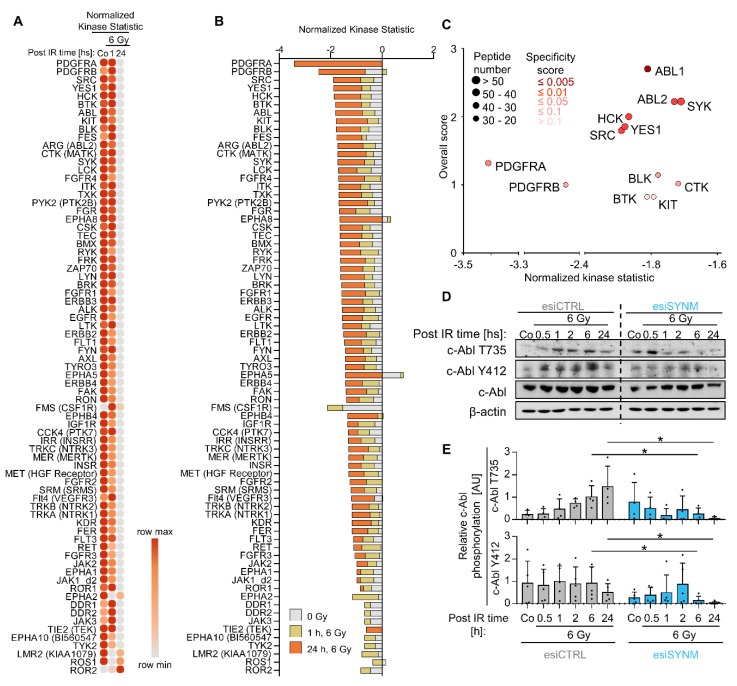
Synemin regulates c-Abl kinase activity. (**A**) Heatmap of tyrosine kinase activities in cells depleted of synemin and exposed to 6 Gy X-ray. Kinase activity profiles were generated by PamGene Technology. (**B**) Normalized kinase statistic of un-irradiated controls (grey) and samples 1 h (yellow) or 24 h (orange) post 6 Gy X-ray exposure. (**C**) Top affected tyrosine kinases in synemin-depleted cell cultures 24 h post 6 Gy X-ray irradiation. *X*-axis indicates the τ value for each kinase (τ < 0 indicates reduced kinase activity relative to control). Dot colors indicate the specificity score. Dot size indicates the number of included peptides. The *Y*-axis shows the overall score. (**D**) Western blotting of c-Abl expression and phosphorylation from whole cell lysates of synemin-depleted cells. (**E**) Densitometries of Western blots from synemin-depleted, 6-Gy irradiated cells showing phosphorylated forms of c-Abl shown in “D” (Y412 and T735, *n* = 4). Phosphorylation levels were calculated relative to the total amount of c-Abl. Dots represent each independent experiment. Data are presented as mean ± SD (two-sided *t*-test; * *p* < 0.05).

**Figure 4 ijms-21-07277-f004:**
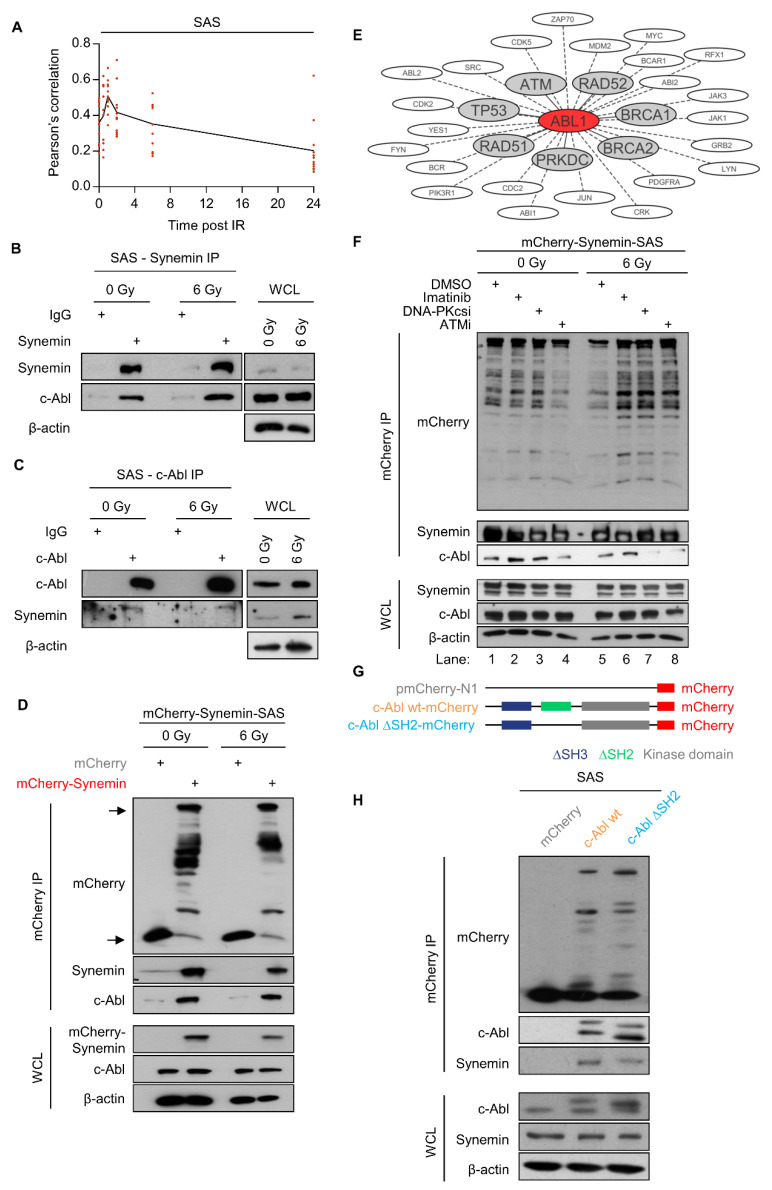
Synemin/c-Abl interact and co-control DSB repair and radiation survival. (**A**) Pearson’s correlation calculated from SAS cells co-stained for synemin and c-Abl at different time point post X-ray exposure (see Appendix A) by using Fiji software. (**B**) Immunoprecipitation of synemin 1 h post 6 Gy X-rays. Western blots show expression of synemin and c-Abl. (**C**) IP of c-Abl 1 h post 6 Gy X-rays. Western blots show expression of synemin and c-Abl. (**D**) Western blots on mCherry immunoprecipitates from 6-Gy irradiated mCherry-SAS and mCherry-Synemin-SAS cells at 1 h post irradiation. β-actin served as loading control. (**E**) c-Abl interactome of DNA repair proteins calculated by Cytoscape. (**F**) Western blots on mCherry immunoprecipitates from mCherry-Synemin-SAS cells after a 1-h pretreatment with Imatinib, DNA-PKcsi or ATMi alone or in combination with 6 Gy X-rays. (**G**) Design of different c-Abl constructs. (**H**) IP of SAS cells expressing mCherry, c-Abl WT-mCherry, and c-Abl ΔSH2-mCherry. IP, immunoprecipitation; WCL, whole cell lysate.

**Figure 5 ijms-21-07277-f005:**
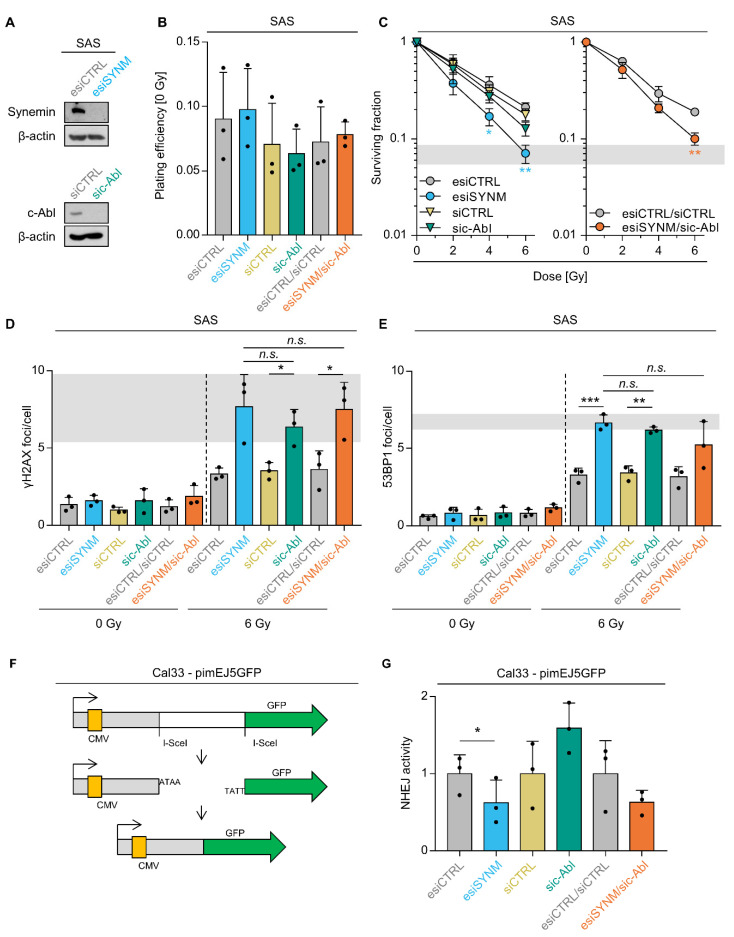
Reduced activity of DNA repair and radiosensitization after Synemin/c-Abl knockdown. (**A**) Knockdown efficiencies of indicated control and specific esi/siRNA in whole cell lysates from SAS cells. β-actin served as loading control. (**B**) Plating efficiency of SAS cells upon single or double knockdown of synemin and c-Abl. Dots represent the mean of each independent experiment. (**C**) 3D clonogenic radiation survival upon single or double silencing of synemin and c-Abl. (**D**) Residual γH2AX and (**E**) residual 53BP1 foci per cell upon single or double knockdown of synemin and c-Abl in 6 Gy X-ray irradiated SAS cells. Transfection with single or double non-specific siRNA was used as controls. Dots represent the mean of each independent experiment. (**F**) Schematic structure of the NHEJ substrate pimEJ5GFP. The pimEJ5GFP has an insertion between the CMV promoter and GFP preventing its translation. Once the endonuclease I-SceI cuts in the proper sites it generates a DSB that can be repair by NHEJ. (**G**) NHEJ activity in Cal33 cells upon single or double knockdown of synemin and c-Abl. Dots represent each independent experiment. Data are represented as mean ± SD (*n* = 3; two-sided *t*-test; * *p* < 0.05, ** *p* < 0.01, *** *p* < 0.001; n.s., not significant (*p* ≥ 0.05)).

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
