# Peer review of "c-Abl Tyrosine Kinase Is Regulated Downstream of the Cytoskeletal Protein Synemin in Head and Neck Squamous Cell Carcinoma Radioresistance and DNA Repair"

_ijms, 2020, doi:10.3390/ijms21197277_

Round 1

Reviewer 1 Report

Authors showed c-Abl and synemin have key role for radiosensitivity in HNSCC cells in 3D culture. Additionally, they showed the connection with NHEJ and provided interesting zebrafish data and molecular kinase data.

It was very interesting and unique for me and may be readers. I enjoyed this manuscript even moderate radiosensitization by knockdown but I think this paper can be improved before publication. 

with figures, esiCTRL is difficult to see. Please change font color from light gray to something else. Maybe darker.

Please make esiRNA (endogenous siRNA) when it shows up the first time.

I am little confused figure 1h-L foci analysis. I am not sure each dot means. M&M did not state how many cells were scored. But three independent experiments were carried out. Did authors conduct stats with total distribution? or with three independent experiments? I am very surprised to see statistical significance in Fig1H-L after irradiation with such a large error bars (SD). 

Authors stated two tailed t-test (paired or unpaired?)and one way anova test were conducted for zebrafish. But presented results were comparison of more than 2. Please specify which test was conducted in each figure legend. I don't know why authors use t-test.

It was unclear for me why si-cAbl was not sensitized in colony formation but leading more foci. Authors used up to 6 Gy for colony formation. If authors use 8 Gy or 10 Gy, greater differences would be appeared and can obtain D10 values, the dose to cause 10% survival. Additionally, why si-cAbl has higher NHEJ activity? It was not stated as statistical significance but it is opposite or suppression of synamin.

Author Response

Comments to reviewer 1:

Authors showed c-Abl and synemin have key role for radiosensitivity in HNSCC cells in 3D culture. Additionally, they showed the connection with NHEJ and provided interesting zebrafish data and molecular kinase data.

It was very interesting and unique for me and may be readers. I enjoyed this manuscript even moderate radiosensitization by knockdown but I think this paper can be improved before publication.

with figures, esiCTRL is difficult to see. Please change font color from light gray to something else. Maybe darker.

Thank you for your suggestion, a darker grey has been used.

Please make esiRNA (endogenous siRNA) when it shows up the first time.

Thank you, weaddeditin the Material and Methods section, but in this case esiRNA stands for endoribonuclease-prepared siRNA.

I am little confused figure 1h-L foci analysis. I am not sure each dot means. M&M did not state how many cells were scored. But three independent experiments were carried out. Did authors conduct stats with total distribution? or with three independent experiments? I am very surprised to see statistical significance in Fig1H-L after irradiation with such a large error bars (SD).

Thank you for your comment. We have performed 3 independent experiments and per each independent experiment 50 randomly picked cells have been quantified per condition. Meaning that in total 150 cells were analysed in the overall experiment. For clarification, this has been added in the figure legend.

Regarding the statistical analysis, we would like to apologize for a mistake. We have performed one-way ANOVA followed by post-hoc analysis using Tukey’s correction of the pooled data, not t-test.

Regarding the standard deviation, foci numbers are usually not normal distributed. However, we can show if we do the mean of the three independent experimentsthat the deviation is small between the experiments (graph attached in PDF).

Authors stated two tailed t-test (paired or unpaired?)and one way anova test were conducted for zebrafish. But presented results were comparison of more than 2. Please specify which test was conducted in each figure legend. I don't know why authors use t-test.

Thank you for your comment.Wethink here is a misunderstanding. T-test was not performedin this case. For zebrafish experiments, one-way ANOVA followed by post-hoc analysis using Tukey’s correction was used for the tail curvature and body length. Instead, for the edema counts, since it contains yes/no parameters, we used Logistic regression analysis.

It was unclear for me why si-cAbl was not sensitized in colony formation but leading more foci. Authors used up to 6 Gy for colony formation. If authors use 8 Gy or 10 Gy, greater differences would be appeared and can obtain D10 values, the dose to cause 10% survival. Additionally, why si-cAbl has higher NHEJ activity? It was not stated as statistical significance but it is opposite or suppression of synamin.

Thank you. Indeed, we see an increase of NHEJ for c-Abl. We interpret this as a compensation for the HR pathway, which is reduced as you can see from the reporter assay in Cal33-DR (confidential information for review only). We did not display these data as no further investigations have been performed on HR. This will be part of future experiments.

(graph attached in PDF)

Reviewer 2 Report

Review Comments:

The Manuscript on c-Abl tyrosine kinase is regulated downstream of the cytoskeletal protein synemin in head and neck squamous cell carcinoma radioresistance and DNA repair” by Sara Sofia Deville et al., demonstrates the involvement of c-Abl tyrosine kinase in the downstream of the cytoskeletal protein synemin in head and neck squamous cell carcinoma radioresistance and DNA repair very well.

I noticed that all methods and results have been presented appropriately to show how of c-Abl tyrosine kinase regulates downstream of the cytoskeletal protein synemin signaling in head and neck squamous cell carcinoma.

However, there are some minor comments in this manuscript

  1. The style of English language used in the manuscript are not easy follow, so if possible, the language could be rephrased.
  2. The labels in the figures should be legible. For example, “esiCTRL” is not readable.
  3. Include densitometry graph for the western blot that shown in Fig.4.

Author Response

Comments to reviewer 2:

The Manuscript on c-Abl tyrosine kinase is regulated downstream of the cytoskeletal protein synemin in head and neck squamous cell carcinoma radioresistanceand DNA repair” by Sara Sofia Deville et al., demonstrates the involvement of c-Abl tyrosine kinase in the downstream of the cytoskeletal protein synemin in head and neck squamous cell carcinoma radioresistance and DNA repair very well.

I noticed that allmethods and results have been presented appropriately to show how of c-Abl tyrosine kinase regulates downstream of the cytoskeletal protein synemin signaling in head and neck squamous cell carcinoma.

However, there are some minor comments in this manuscript

1.The style of English language used in the manuscript are not easy follow, so if possible, the language could be rephrased.

Thank you. We improved some parts accordingly. The corrections are seen in tracked mode.

2.The labels in the figures should be legible. For example, “esiCTRL” is not readable.

Thank you for your suggestion, a darker grey has been used.

3.Include densitometry graph for the western blot that shown in Fig.4.

Thank you for your comment. We have added the densitometries in Figure S4A-C.